# Effect of Six Weeks of Training with Wearable Resistance Attached to the Forearm on Throwing Kinematics, Strength, and Velocity in Female Handball Players

**DOI:** 10.3390/jfmk10010045

**Published:** 2025-01-24

**Authors:** Andrea Bao Fredriksen, Roland van den Tillaar

**Affiliations:** Department of Sports Sciences and Physical Education, Nord University, 7600 Levanger, Norway; andreabaof@hotmail.com

**Keywords:** forearm loading, strength training, team handball, overarm throwing, velocity

## Abstract

**Background:** The aim of the study was to investigate the effect of six weeks of training with wearable resistance attached to the forearm on throwing kinematics, strength, and velocity in experienced female handball players. **Methods:** Twenty-five female handball players (mean age: 24.7 ± 3.8 y, mass: 72.1 ± 17.6 kg, height: 1.69 ± 0.06 m, and training experience: 14.3 ± 4.9 y) participated and were divided into an experimental group (*n* = 14) or a control group (*n* = 11). Both groups participated in the same handball sessions, but the experimental group trained with wearable resistance attached to the forearm during the training sessions. Before and after a training period of six weeks, throwing velocity, strength, and kinematics were tested. **Results:** The throwing velocity was significantly increased in the experimental group but not in the control group (*p* = 0.006). Only a few significant kinematic changes were observed, mostly around ball release, in which both groups had increased elbow flexion. The experimental group had a larger shoulder flexion at ball release and a more flexed wrist (*p* ≤ 0.04). Meanwhile, the control group had a lower shoulder abduction after the training period (*p* ≤ 0.023). No significant effects were found in the maximal angular velocities, timing of joint angles, angular velocities and in the strength test (*p* > 0.075). **Conclusions:** Based on the findings in this present study, it was concluded that training with wearable resistance attached to the forearm increases throwing velocity during the competition season compared to normal throwing training, which was probably caused by the changes in maximal elbow angle and extension velocity and the more flexed wrist at ball release.

## 1. Introduction

In team handball, such as javelin throw, baseball, cricket, and water polo, throwing is an important performance factor [1,2]. In team handball, throwing is an essential technique to pass and shoot the ball into the opponent’s goal to score points. The throwing technique can be measured with two different performance parameters, throwing velocity and accuracy [3,4]. These performance parameters are influenced by throwing kinematics [5]. Several studies have investigated throwing kinematics upon these performance parameters in team handball [5,6,7,8,9,10,11]. van den Tillaar and Ettema [12] found out that elbow extension and internal shoulder rotation are the main contributors in overarm throwing in team handball, and 73% of the ball velocity can be explained by that. Therefore, it was suggested that improving the throwing technique through optimizing the timing of consecutive actions of body segments [13,14] and arm muscle strength in the specific muscles for internal shoulder rotation and elbow extension could increase throwing velocity. Furthermore, other studies have examined the strength of the upper limb, and how this influences the throwing velocity [1,13,15,16,17,18,19,20,21,22].

To gain throwing velocity, several strength training programs are used in training, which can be divided into two groups: general vs. specific strength training [23]. As general strength training refers to training the capacity of the whole muscular system in a more general way, specific strength training refers to similar movements to the actual sport in movement and velocity. In a review, van den Tillaar [24] showed that both specific training programs with underweight balls and general strength training programs with heavy weights could positively influence throwing velocity. A recent review [25] investigated the effect of only specific strength training methods and found that, besides throwing underweighted balls, resistance band-specific strength training methods were the most effective methods to ensure an increase in throwing velocity. Training throws with a pulley device (heavy loads), throwing overweight balls, and forearm loading resulted in more conflicting results [25]. However, there are some challenges with the different training methods. There are underweight balls of a similar size as regular balls; with the overweighted balls, it is not possible to individualize the weights and throwing with a resistance band and pulley device cannot be conducted during regular throwing practice in training. Yet, forearm loading is easy to implement during throwing practice during a handball training session, and the load can be easily individually adjusted.

However, to our best knowledge, only two studies investigated the wearing of weights attached to the forearm (specific strength training) upon throwing [16,26]. They used, roughly, a 0.1 kg load on the forearm and upper arm and found an increase in both the experimental group and the control group, which could be explained by the fact that the participants were novice handball players. Thus, the results for novice players might be misleading due to a learning effect during the study and no training effect. Thus, these results should not be generalized for experienced handball players. Furthermore, no kinematic analysis was performed that could explain the changes in throwing velocity after the intervention period.

Therefore, the aim of the present study was to investigate the effect of six weeks of training with wearable resistance attached to the forearm on throwing kinematics, strength, and velocity in female handball players. It was hypothesized that, due to the gradual overload of forearm loading, while keeping the same throwing technique, throwing strength will increase, and thereby the throwing velocity over time will increase.

## 2. Methods

### 2.1. Experimental Design and Methodology

A pre-test–post-test randomized controlled study was used to examine the training effect of six weeks of training with wearable resistance attached to the forearm in female handball players. The participants were randomly assigned to an experimental group that trained their regular handball sessions with external weights attached to their forearms, while the other half was assigned to a control group that continued their regular handball practice without external weights. Both groups conducted the same training and had two handball practices per week. Pre- and post-tests were conducted, and changes in maximum throwing velocity, joint angles, angular velocities, ball release kinematics, as well as a pulley-device strength test were compared. The inclusion criteria were (a) females, 16–40 years old; (b) a minimum of five years of handball experience; and (c) playing handball actively during the period of the study and the year before. The exclusion criteria were (a) injuries preventing maximum effort in throwing velocity and (b) failure to attend more than 85% of the training sessions. All of the testing and training were performed in the competitive in-season (September–December and January–March).

### 2.2. Participants

A minimum sample size of 12 for each group would provide a power of 0.80, calculated by using G*power (Version 3.1.9.6, University of Dusseldorf, Dusseldorf, Germany) [27]. The power analysis was computed with an effect size of 0.6 and an alpha level of 0.05. Therefore, a sample of more than 12 players in each group was assembled. At the start of this study, 30 senior female handball players volunteered. They were randomly assigned to an experimental wearable resistance training group (*n* = 15) and a control group (*n* = 15). The groups were balanced for player positions and throwing performance at the pre-test. During the study, five subjects were excluded due to injuries (two subjects in the control group) not related to the study, and three subjects were excluded due to them not meeting the attendance requirement of 85% (two subjects in the control group and one subject in the experimental group). Therefore, a total of 25 (experimental = 14, control = 11) female handball players (mean age: 24.7 ± 3.8 yrs, mass: 72.1 ± 17.6 kg, height: 1.69 ± 0.06 m, and training experience: 14.3 ± 4.9 yrs) completed the study. They played in the fifth division of the national competition. The participants were fully informed about the procedures, the potential risks, and the benefits of the study in written and oral form. Written consent was obtained before all testing. Parental consent was obtained for the subjects < 18 years. The study was conducted following the latest revision of the Declaration of Helsinki and approved by the local ethics committee and the Norwegian Centre for Research Data (project number 189749; approval date: 19 April 2022).

### 2.3. Procedure

All testing procedures took place three to seven days before and after the six-week training intervention. Pre- and post-testing was completed on a single day, with a general warm-up of a minimum of 15 min, including throwing drills. The participants’ throwing performance was tested 5 m away from a square target (0.6 × 0.6 m) at a height of 1.5 m in the middle of a handball goal (2 × 3 m). They had to perform a standing throw, with the front foot always standing on the ground like a penalty shot (Figure 1). For the left-handers, everything was mirrored. The participants were instructed to throw as hard as possible and try to hit the target, and they had to throw until they performed five approved throws (hitting the target while throwing as hard as possible). Approximately one minute of rest between each throw was provided to avoid fatigue.

After throwing, test participants had to perform a strength test to investigate if a possible change in throwing velocity was caused by increased strength. For this strength test, a warm-up set on the pulley device machine was conducted before starting throwing testing. The warm-up set on the pulley device was used to familiarize participants with the pulley device machine before testing to prevent a learning effect. The strength test was an overarm throwing movement while pulling a pulley device with different loads to build a load–velocity relationship to measure the overarm throwing strength. The subjects had to stand still with the front foot in the same position as in the penalty shot, and the height was adjusted to the subject, so the pulling movement was performed in the same plane as throwing normal throws, according to the same method as described by Ettema et al. [13] (Figure 2). A single grip (Redcord AS, Arendal, Norway) was connected to the pulley device, and the subjects were instructed to throw normally as fast as possible. Three repetitions per load with three different loads (5–15 kg) were tested, and a load–velocity linear regression was performed to investigate the relationship. The pause between repetitions was 30–45 s, and the pause between each weight was approximately two minutes.

### 2.4. Measurements

The peak throwing velocity was measured using a radar gun (Stalker ATS II, Richardson, TX, USA) with ±0.028 m/s accuracy within a field of 10 degrees from behind the participant [28]. For kinematic data, eight three-dimensional motion capture cameras (Qualisys, Gothenburg, Sweden) operating at a frequency of 240 Hz were used to measure the position of reflective markers (14 mm) on the following anatomical landmarks on the throwing arm: (a) finger, middle distal phalanx; (b) hand, middle metacarpal head; (c) wrist, radial and ulnar styloid process; (d) elbow, lateral and medial epicondyles; (e) shoulder, lateral tip of the acromion on both sides; and (f) thorax, sternum. Furthermore, reflective markers were attached on the opposite foot of the throwing arm: (a) hip, trochanter major on both sides; (b) pelvis, posterior superior iliac spine and anterior superior iliac spine on both sides; (c) knee, lateral and medial epicondyles; (d) ankle, lateral and medial malleolus; and (e) foot, first and fifth proximal phalanx. After the warm-up, these reflective markers were attached. The ball had four reflective markers on the top, left, right, and in front. The time of ball release was derived when the distance between the finger marker and the top marker on the ball increased intensively and abruptly [1,5,6].

All of the kinematic data were exported to C3D files for segmentation modeling and the subsequent analysis in Visual3D (C-motion, Germantown, MD, USA). The model-based data were smoothed with a low-pass Butterworth filter with a cut-off frequency of 20 Hz. All joint angles (Figure 3) were defined as the proximal segment relative to the distal segment with a Cardan sequence in the order x-y-z, except for the shoulder joint angle, which was with a Cardan sequence of z-y-z based on the ISB recommendation [29]. The angles and angular velocities for the wrist, elbow, shoulder, and trunk were calculated at ball release, as well as the maximum angles and angular velocity before ball release. The timings for the different kinematic parameters were calculated, and ball release was set to time 0, and the time before ball release was defined as negative [1].

The pulley device test was conducted to investigate if the effect of external weights on throwing velocity could be explained by the changes in strength. Therefore, a strength test with a throwing-like movement was conducted to measure strength in the throwing arm and to make a load–velocity profile. A load–velocity profile shows the relationship between strength and velocity, and with these data, we can estimate the one-repetition maximum (1RM) without an actual maximal test [30,31]. To establish this load–velocity relationship on the strength data from the pulley device, the average velocity was measured using a linear encoder (Ergotest Technology AS, Langesund, Norway) to load the pulley device. Three attempts were made with three different weights, and the mean of the two best average velocities on each weight was used for the analyses to calculate the 1RM. In a pilot study, the 1RM was tested and showed that the average velocity for 1RM in this pully device is approximately 0.5 m/s. The load–velocity profiles were based on a linear regression on the average concentric velocity on each load. To obtain the estimated 1RM, the linear regression equation (y=mx+b) was used. This equation describes the slope–intercept form of a straight line, where ‘m’ describes the slope of the line, ‘*x*’ is the x-axis value, and ‘*b*’ is the point where the line intersects with the y-axis. The ‘*m*’ and ‘*b*’ were used for statistical analyses to detect differences within groups and between groups.

### 2.5. Training Intervention

This study was performed twice in different periods, September to December and January to March (a six-week period), with the same procedure each time due to the lack of participants the first time and the availability of equipment and participants. Both training periods took place during the competitive season, and subjects from both the experimental and control groups participated in the competition during the training intervention period. Twelve training sessions were performed over six weeks, and approximately one competition was held per week. Between each training session was at least 48 h of rest. All of the subjects participated in the same handball training sessions led by the coach. All of the training sessions lasted 90 min. The training sessions consisted of a general and specific warm-up, throwing drills, shooting, handball tactics, running, sprinting, and handball playing. Both groups participated in the same sessions, but the experimental group used external weights attached to the forearm (Figure 4) by an exogen forearm sleeve (Lila™, Kuala Lumpur, Malaysia). They used forearm sleeves and weights on both arms. In the first week, the experimental group started with 0.1 kg to gradually increase the load since an earlier study has shown that a steep increase in forearm loading changes the technique [1]. The weights progressively increased from week to week by 0.05 kg, and in the last week (week six), the weights were the same as in week five (Table 1) due to tapering before post-testing [32]. The chosen weights are based on earlier studies with overweight balls [33]. In the final week, the attached load on the forearms was approximately 86–92% heavier than the regular women’s handball ball.

### 2.6. Statistics

The descriptive statistics were presented as means and standard deviation (SD), and the data were checked for normality distribution using the Shapiro–Wilk test. The training effect of external weights on the forearm was analyzed using a two-way analysis of variance (ANOVA) with repeated measures 2 (time: pre-, post-test) × 2 (group: experimental vs. control group) on maximal throwing velocity, strength pulley device test, joint angles at ball release, maximal joint angles and maximal joint angular velocities before ball release, and their timing. When significant differences were observed, post hoc comparisons were conducted to find out where the differences were. The assumption of sphericity was controlled with Mauchly’s test of sphericity. If sphericity was violated, Greenhouse–Geisser corrections were reported. The significance level was set at *p* < 0.05 to identify differences. The effect size (ES) was evaluated with partial Eta squared (η_p_^2^) where <0.01–0.06 η_p_^2^ constitutes a small effect, <0.06–0.14 η_p_^2^ constitutes a medium effect, and >0.14 η_p_^2^ constitutes a large effect [34]. Statistical analyses were conducted using IBM SPSS 28.0 (IBM Corp., Armonk, NY, USA).

## 3. Results

### 3.1. Velocity

No significant effect of test occasion (F = 1.46, *p* = 0.2, η_p_^2^ = 0.06) and group (F ≤ 4.08, *p* ≥ 0.055, η_p_^2^ ≤ 0.15) was found on throwing velocity. However, a significant interaction effect (test time*group) was found (F = 9.33, *p* = 0.006, η_p_^2^ = 0.29). Post hoc comparisons revealed a significantly higher throwing velocity in the experimental group from pre- to post-test (*p* = 0.004, η_p_^2^ = 0.31, Figure 5), while no significant changes were observed in the control group (*p* = 0.23, η_p_^2^ = 0.06, Figure 5).

### 3.2. Kinematics

A significant effect of test occasion was found for the maximal shoulder flexion (F = 7.76, *p* = 0.01, η_p_^2^ = 0.25), elbow flexion (F = 5.77, *p* = 0.025, η_p_^2^ = 0.2), and trunk extension angles (F = 5.74, *p* = 0.025, η_p_^2^ = 0.2). However, no significant group (F ≤ 3.85, *p* ≥ 0.062, η_p_^2^ ≤ 0.14) and interaction effects (F ≤ 2.2, *p* ≥ 0.15, η_p_^2^ ≤ 0.09, Table 2) were found for the maximal angles. Post hoc comparisons revealed that only the experimental group had a significantly larger shoulder flexion angle in the post-test (*p* = 0.022). Meanwhile, only the control group had a larger trunk extension from pre- to post-test (*p* = 0.016). The maximal elbow flexion angle was significantly more flexed from pre- to post-test when both groups were combined but did not reach a significance level for any of the two groups individually (*p* ≥ 0.076), Table 2). Furthermore, no significant effects were found in timing for maximal angles (F ≤ 2.63, *p* ≥ 0.11, η_p_^2^ ≤ 0.1, Table 2).

At ball release, a significant effect of test occasion was observed in the wrist flexion, elbow flexion, shoulder abduction, and shoulder flexion (F ≥ 4.71, *p* ≤ 0.04, η_p_^2^ ≥ 0.17, Table 2) angles. However, no significant group nor interaction effects (F ≤ 1.89, *p* ≥ 0.18, η_p_^2^ ≤ 0.08, Table 2) were observed for any of the joint angles at ball release. Post hoc comparisons revealed a significantly more flexed elbow and wrist and shoulder flexion angle at ball release at post-test for the experimental group (*p* ≤ 0.043) and only a more flexed elbow and less shoulder abduction in the control group (*p* = 0.023, Table 2).

### 3.3. Maximal Angular Velocity

No significant effects were found in the maximal angular velocities (F ≤ 3.49, *p* ≥ 0.075, η_p_^2^ ≤ 0.13, Table 3). However, a medium interaction effect size (test time × group) was found in the maximal elbow extension velocity (F = 3.49, *p* = 0.075, η_p_^2^ = 0.13, Table 4). Furthermore, a medium effect size of shoulder abduction was found on the test occasion (F = 3.41, *p* = 0.078, η_p_^2^ = 0.12). No significant differences were found in the timing of maximal angular velocities (F ≤ 2.97, *p* ≥ 0.1, η_p_^2^ ≤ 0.12, Table 4).

### 3.4. Pulley Device Test

No significant effects were found for the slope, intercept and estimated 1RM in the pulley device test (F ≤ 2.92, *p* ≥ 0.1, η_p_^2^ ≤ 0.11, Table 5). However, a medium effect size in the estimated 1RM was found for the control group from pre- to post-test (*p* = 0.075, η_p_^2^ = 0.13) due to the increase for all subjects from pre- to post-test except for one, while in the experimental group, the results varied (Figure 6).

## 4. Discussion

The purpose of the study was to investigate the six-week training effect of wearable resistance on the forearm on throwing velocity, kinematics, and strength. The main findings were that the experimental group had an increase in throwing velocity, whereas the control group decreased in throwing velocity. Furthermore, some small adaptations to throwing kinematics were found after the intervention period in both groups. However, no significant difference between groups and interaction effects were observed in the kinematics or strength test.

The reported maximal ball velocity in this study was in the same range (18 m/s) as in earlier studies on experienced female handball players [1,15,35], while being lower than those at the elite level [10,36]. This indicates that the throwing performances are at a respectable level but not at the highest level. The ball velocity in the experimental group increased significantly to around 0.67 m/s (4%), while that in the control group did not change after the training period. This percentage of increase was in line with the previous studies (2.8–3.7%) that used underweighted, overweighted, or a combination of these balls in throwing training [33,37,38,39]. Yet, the results were much lower than in the previous studies on forearm loading [16,26] that reported increases of around 12.3–13%. This discrepancy can be explained by the level of participants used, as in the previous studies, novice players were used, while in the present study, they were experienced. In the previous studies, an increase of 6% was also found in the control group, indicating a general learning effect. When comparing the two training groups with each other in the study by Kotzamanidis et al. [26], the difference was only 5.3, which is more comparable to the findings between the groups in the present study.

The higher ball velocity is probably caused by more efficient throwing kinematics, as shown in previous studies between different skilled throwers [40,41]. However, in the present study, not many clear changes in joint kinematics were observed that could explain the increase in ball velocity. Notably, maximal shoulder flexion and shoulder flexion at ball release did increase after the intervention period in the experimental group. The increased moment of inertia of the extra loads attached to the forearm probably takes more force to accelerate during the acceleration phase but also takes more energy to decelerate in the follow-through phase of the throw. As a result, over time, the shoulder flexion angle increases, but a larger maximal elbow flexion and flexion at ball release also occurs, as observed in the study. A more flexed maximal elbow angle gives a longer distance from the axis of rotation. In that way, it is possible to create a larger torque, which creates more power and results in higher velocity [42]. Furthermore, the wrist was more flexed after the intervention period in the experimental group, which also, again, potentially creates a longer trajectory to produce force to the ball. These differences in angles did not result in significant differences in maximal joint angular velocities. However, a medium effect size was found for the interaction time group on maximal elbow extension velocity, indicating that both groups had another development from pre-to post-test: the maximal angular elbow extension velocity in the experimental group increased, while that in the control group decreased, although not significantly.

It was hypothesized that the experimental group would increase in strength due to the wearable resistance on the forearm, with no expected increase in the control group. However, the strength test did not reveal significant differences between the groups. This discrepancy suggests that the increased ball velocity may be due to neural adaptations, which are known to occur earlier than muscular strength gains [43]. The improvement in throwing velocity in the experimental group could result from the better coordination and activation of the relevant muscles for this specific movement [44]. Additionally, it could be attributed to the increased activation of motor units and a higher firing rate [45]. The relatively short training period or insufficient load variation might also have limited improvements in the specific strength test. However, electromyography (EMG) measurements were not included in this study, which could verify these suggestions.

Moreover, the strength test measured the overall throwing movement rather than isolating specific muscle groups. With an increased load on the forearm, it is likely that muscles, including the flexors, pronators, and grip muscles, take on the majority of the additional strain, potentially resulting in strength gains in these specific areas [46]. Since the strength test tested the entire throwing movement with rather heavy loads (5–15 kg), this affects the throwing movement greatly and probably targets other large muscle groups, like the core and trunk muscles, much more than the muscles involved with forearm loading with some small extra loads.

This study has some limitations. Other strength tests should be performed that are more suitable for testing only the upper body throwing movement instead of the present strength test, as the current test probably did not target the right muscles enough. However, the advantage of this strength test was that it mimics the throwing movement, which is the most specific strength test for overhead throwing [47]. Another challenge was in recruiting a sufficient number of participants who met the inclusion criteria. The findings are, therefore, applicable only to experienced intermediate-level handball players. The advantage of using these participants was that both groups were from the same training group and thereby, the training content was similar for both, except for the wearable resistance. It would be valuable to examine forearm loading in elite handball players before the findings are generalized to higher levels. Additionally, the six-week training intervention may have been too short to observe significant strength gains from throwing training with external weights. In the present study, all of the players followed the same load increase every week, which could influence the results as some players were stronger, taller, and heavier than others. Therefore, the training stimulus could be different for each player. Perhaps individualizing the forearm load based on anthropometrics and/or upper body strength could overcome this limitation, which is easy to alter by adjusting the weight of the wearable resistance attached to the sleeve during training.

Moreover, the present study only analyzed discrete kinematic data points that occur during the throwing movement (maximal angles/angular velocity and ball release), which does not give all of the information due to the throwing technique variations within and between players. Later studies should include the analysis of the entire throwing technique by using analyses like statistical parametric mapping [48,49] or neural networks [50] that could investigate this. Finally, electromyography should also be included to investigate the long-term effects of training with wearable resistance on muscle activation, which could give more information about the possible adaptations of the muscles involved in overarm throwing over time.

## 5. Conclusions

Based on the findings in this study, it can be concluded that training with wearable resistance attached to the forearm can increase throwing velocity during the competition season, compared to normal throwing training with regular balls. The findings in the present study suggest that the increase in throwing velocity might be due to the changes in maximal elbow angle, increased elbow extension velocity and the more flexed wrist at the ball release. However, these results are based on experienced intermediate-level handball players, so they cannot be generalized to elite players. The advantage of using forearm loading in practice, for trainers and players, is that it can easily be incorporated throughout an entire training session, including various throwing drills and game sequences as, unlike throwing and shooting with overweighted balls, players can safely shoot at the goalkeeper without increasing the risks of injury. Furthermore, the weights can be individualized based on the player’s position and physical capacity, allowing for tailored training programs. This approach enables players to progress at their level while adhering to the principles of progressive overload, ensuring positive adaptations over time.

## Figures and Tables

**Figure 1 jfmk-10-00045-f001:**
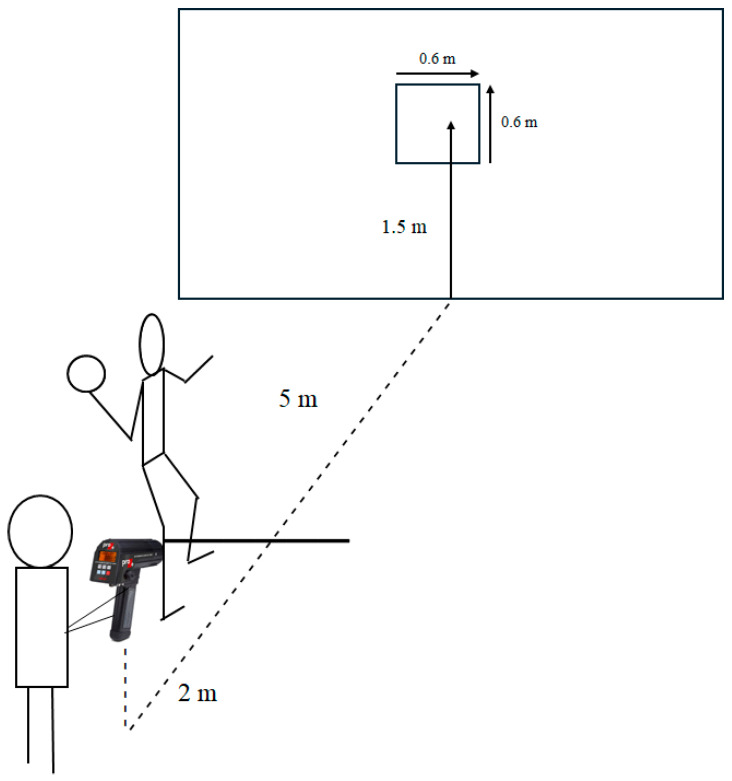
Illustration of the experimental setup. Subjects throwing on a target drawn on a large mattress 5 m away. Velocity was measured by a radar gun from behind the subjects.

**Figure 2 jfmk-10-00045-f002:**
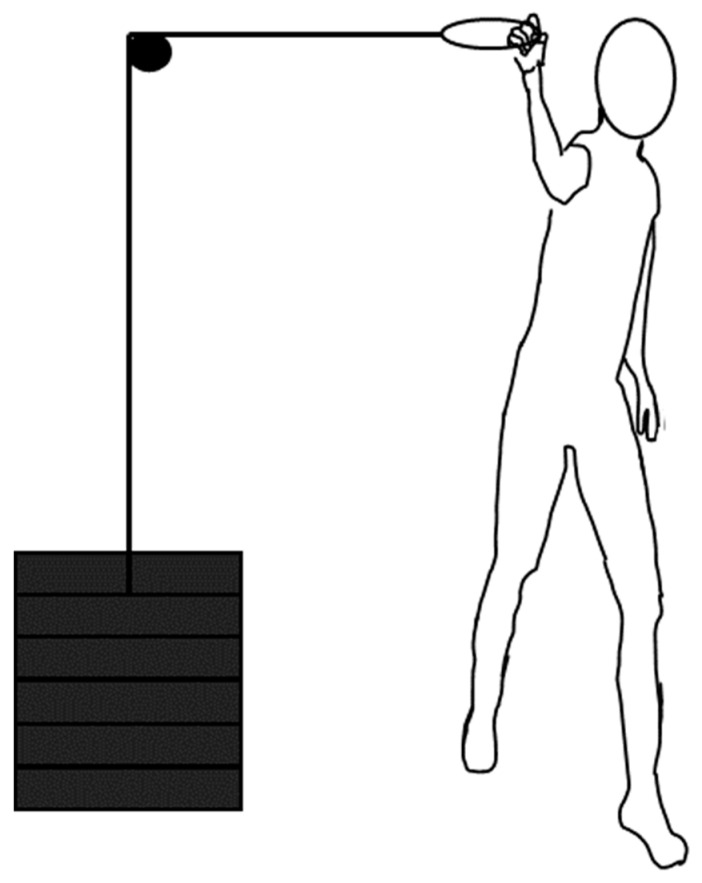
Illustration of the pulley device test.

**Figure 3 jfmk-10-00045-f003:**
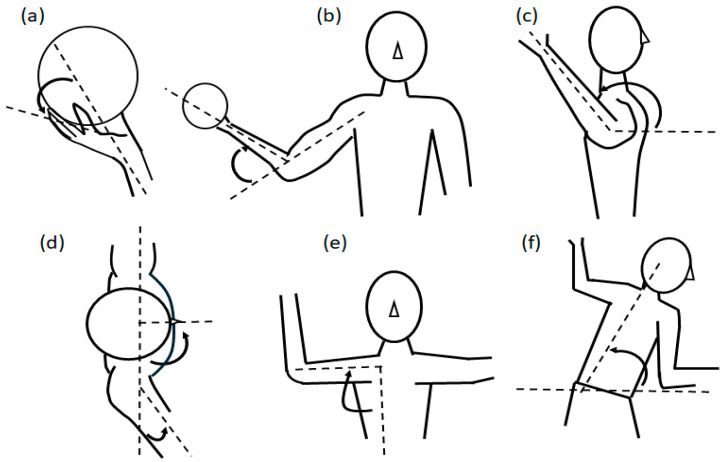
Definition of the different kinematic parameters (**a**) wrist extension, (**b**) elbow flexion, (**c**) shoulder internal rotation, (**d**) shoulder flexion and trunk rotation, (**e**) shoulder abduction, and (**f**) trunk flexion.

**Figure 4 jfmk-10-00045-f004:**
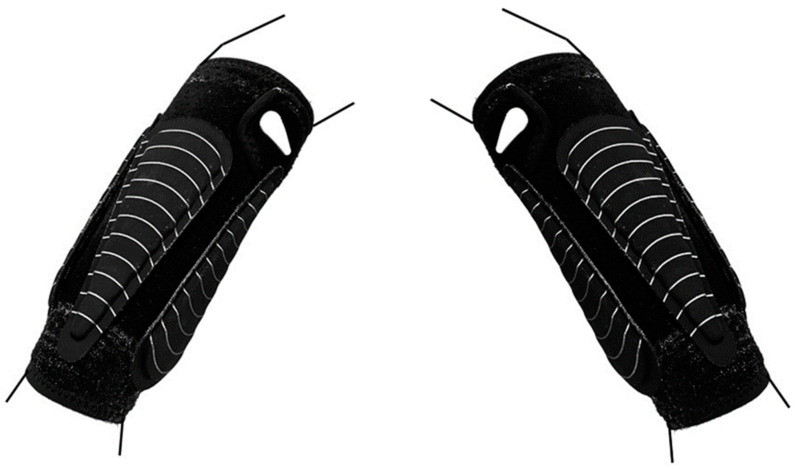
The forearm sleeves and the attachment of the weights. Adapted from Fredriksen and van den Tillaar [1].

**Figure 5 jfmk-10-00045-f005:**
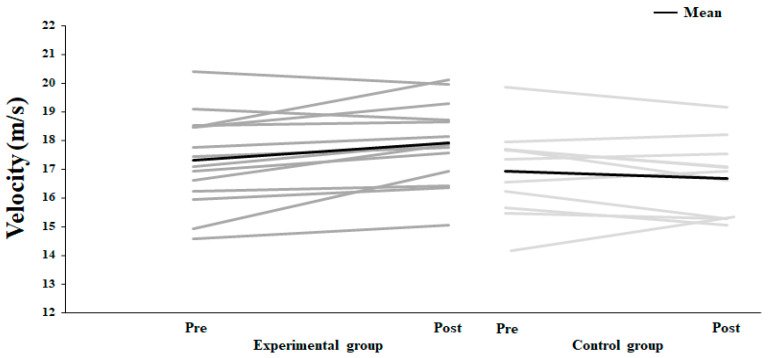
Maximal throwing velocity for each subject (grey lines) and average per group (black line) on pre- and post-test.

**Figure 6 jfmk-10-00045-f006:**
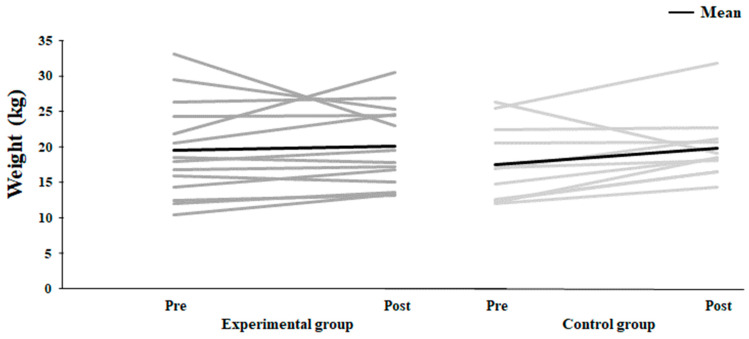
Estimated 1RM (mean ± SD) in the pulley device test for pre- and post-test for each individual (grey line) and average for the experimental and control groups (black line).

**Table 1 jfmk-10-00045-t001:** The different weights placed on the forearm per training session for the experimental group.

Week	0	1		2		3		4		5		6		7
Session	Pre-test	1	2	3	4	5	6	7	8	9	10	11	12	Post-test
Training weight		0.1		0.15		0.2		0.25		0.3		0.3		

**Table 2 jfmk-10-00045-t002:** Maximal angles, angles at ball release (mean ± SD) for pre- and post-test for the experimental and control groups. * indicates a significant difference between pre- and post-test (*p* ≤ 0.05).

	Experimental Group				Control Group			
	Ball Release		Max Angle		Ball Release		Max Angle	
	Pre-Test	Post-Test	Pre-Test	Post-Test	Pre-Test	Post-Test	Pre-Test	Post-Test
Wrist extension	4 ± 13.5	−5.7 ± 1 7 *	32 ± 16.4	39.7 ± 21.6	2 ± 13.9	−1.9 ± 15.9	35 ± 13.5	36.2 ± 12.6
Elbow flexion	53 ± 13.6	58.7 ± 12.1 *	109.5 ± 16.5	112.6 ± 15.8 *	55.9 ± 16.4	61.2 ± 18.7 *	110.8 ± 13.7	113.2 ± 14.8
*Shoulder*								
Internal rotation	101.7 ± 14.2	105.1 ± 15.2	136.9 ± 13.3	135 ± 11.5	97.4 ± 12.5	93.9 ± 18.5	130.2 ± 10	125.3 ± 9.9
Flexion	−20.3 ± 6.4	−25.3 ± 7.3 *	−27.2 ± 8	−30.9 ± 9.1 *	−23.4 ± 7.4	−25.9 ± 8	−29.5 ± 8.2	−31.6 ± 8.7
Extension	-	-	13.2 ± 11.7	10.9 ± 13.8	-	-	14.9 ± 14.6	12 ± 14.7
Abduction	88.2 ± 9.8	87.2 ± 10.2	90.3 ± 8.6	90.8 ± 7.5	85.1 ± 8.3	83.2 ± 8.2 *	87.2 ± 6.7	85.6 ± 7.7
*Trunk*								
Flexion	20.5 ± 3.8	20 ± 4	69.2 ± 3.7	69.9 ± 4	19.5 ± 5.3	17.6 ± 6.2	70.5 ± 5.3	72.2 ± 5.7
Extension	-	-	98.8 ± 7	100.5 ± 4	-	-	98.2 ± 4.6	100.6 ± 4.4 *
Rotation	10.9 ± 9.5	10.1 ± 9.9	36.9 ± 11.7	36 ± 10.6	5 ± 11.5	8 ± 11.2	29.1 ± 10	32 ± 9.8

**Table 3 jfmk-10-00045-t003:** Timing of maximal angles (mean ± SD) for pre- and post-test for the experimental and control groups.

	Experimental Group		Control Group	
	Timing Max Angle			
	Pre-Test	Post-Test	Pre-Test	Post-Test
Wrist extension	−0.129 ± 0.087	−0.110 ± 0.063	−0.132 ± 0.082	−0.109 ± 0.071
Elbow flexion	−0.114 ± 0.080	−0.137 ± 0.133	−0.139 ± 0.119	−0.126 ± 0.068
Shoulder internal rotation	−0.054 ± 0.041	−0.038 ± 0.016	−0.043 ± 0.016	−0.054 ± 0.039
Shoulder flexion	−0.041 ± 0.051	−0.021 ± 0.012	−0.028 ± 0.013	−0.033 ± 0.026
Shoulder extension	−0.304 ± 0.119	−0.291 ± 0.108	−0.349 ± 0.105	−0.350 ± 0.062
Shoulder abduction	−0.055 ± 0.073	−0.057 ± 0.075	−0.030 ± 0.043	−0.063 ± 0.060
Trunk flexion	−0.005 ± 0.015	−0.002 ± 0.005	−0.003 ± 0.007	−0.012 ± 0.026
Trunk extension	−0.364 ± 0.118	−0.351 ± 0.131	−0.355 ± 0.147	−0.302 ± 0.092
Trunk rotation	−0.186 ± 0.041	−0.178 ± 0.034	−0.194 ± 0.038	−0.194 ± 0.051

**Table 4 jfmk-10-00045-t004:** Maximal angular velocities (mean ± SD) and the timing of their max velocity.

	Experimental Group				Control Group			
	Max Angular Velocity		Timing		Max Angular Velocity		Timing	
	Pre-Test	Post-Test	Pre-Test	Post-Test	Pre-Test	Post-Test	Pre-Test	Post-Test
**Wrist flexion**	1329.2 ± 358.7	1315.4 ± 191.6	−0.008 ± 0.012	−0.010 ± 0.013	1284.4 ± 2908	1209.1 ± 184.5	−0.005 ± 0.007	−0.004 ± 0.008
**Elbow extension**	1291.1 ± 305.1	1357.9 ± 305.9	−0.009 ± 0.004	−0.008 ± 0.009	1331.3 ± 185.5	1270 ± 199	−0.013 ± 0.008	−0.006 ± 0.006
**Shoulder internal rotation**	1798.9 ± 656.1	1864.9 ± 781	0	0	1635.9 ± 577.6	1516 ± 768.4	0	0
**Shoulder flexion**	714.9 ± 286.8	737.5 ± 175.3	−0.013 ± 0.029	−0.008 ± 0.015	709.4 ± 196.4	728 ± 273.4	−0.024 ± 0.038	−0.018 ± 0.040
**Shoulder abduction**	433.2 ± 175.3	393.8 ± 173.6	−0.085 ± 0.095	−0.096 ± 0.014	351.8 ± 148.4	310.4 ± 119	−0.123 ± 0.145	−0.123 ± 0.118
**Trunk flexion**	371.4 ± 152.9	367.9 ± 121.8	−0.029 ± 0.035	−0.029 ± 0.022	270.5 ± 64.8	291.6 ± 93.7	−0.021 ± 0.011	−0.018 ± 0.012
**Trunk rotation**	264 ± 61.1	266.4 ± 62.3	−0.079 ± 0.034	−0.079 ± 0.029	238.4 ± 33.7	242.3 ± 68	−0.063 ± 0.029	−0.070 ± 0.019

**Table 5 jfmk-10-00045-t005:** Pre-and post-test results (mean ± SD) of the m (slope) and b (intercept) in the pulley device test.

	Experimental Group		Control Group	
	Pre-Test	Post-Test	Pre-Test	Post-Test
*m*	−9.6 ± 3.9	−7.7 ± 2.9	−8.7 ± 3.7	−9.6 ± 6.0
*b*	24.3 ± 8.5	23.9 ± 6.7	21.9 ± 7.1	24.7 ± 7.3

## Data Availability

The data presented in this study are available upon request from the corresponding author. The data are not publicly available due to the national laws of the Norwegian government regarding privacy.

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
