# Peer review of "Effect of Six Weeks of Training with Wearable Resistance Attached to the Forearm on Throwing Kinematics, Strength, and Velocity in Female Handball Players"

_jfmk, 2025, doi:10.3390/jfmk10010045_

Round 1

Reviewer 1 Report

Comments and Suggestions for Authors

Deear Authors,

Thank you for the opportunity to revise your article, the following suggestions could improve the quality of your work. 

INTRODUCTION

* I would start with a description of the impact of overhead injuries in the shoulder the associated fatigue, then I would introduce handball and throwing. This structure could support better your aim.

*are previous study highlight differences between female and male that could support the choice of your sample? Sex-based differences have been reported in morphological and neuromuscular characteristics of such muscles that could influence also sport injuries but also sport performance. I think that this aspect should be described in order to support your work. 

*" Training throws with a pulley device (heavy loads), throwing overweight balls, and forearm loading resulted in more conflicting results. However, there are some challenges with the different training methods. There are no similar size underweight balls as regular balls; with the over-weighted balls, it is not possible to individualise the weights and throwing with a resistance band and pulley device cannot be conducted during regular throwing practice in  training. Yet, forearm loading is easy to implement during throwing practice during a handball training session, and the load can be easily individually adjusted." references are missing

METHOD

*Inclusion criteria: the range of years is more large had all female examined already had menarche?

*in which phase of menstrual cycle were the participants assessed? Menstrual cycle could modify the sport performance, please specify it.

* 2.4. Measurements. Please add more references that support the chosen instruments of assessment. 

RESULTS

*no comment

DISCUSSION

* the results concerning no significant difference in the kinematics and strength test should be better discuss suggesting general hypotheses. 

the reference to figures and tables in the text should not be placed in the discussions but only in the results, line 308, 318, 323, 327

* in general the discussion need a clear statement of the novelties presented in this study

Author Response

We want to thank the reviewers for their comments to the manuscript. We have tried to answer to all the comments of the reviewers and changes are colored red in the manuscript. We think that the manuscript now is suitable for publication.

Reviewer 1

INTRODUCTION

* I would start with a description of the impact of overhead injuries in the shoulder the associated fatigue, then I would introduce handball and throwing. This structure could support better your aim.

The main aim of the present study was to increase throwing performance in handball. Thereby, we don’t think that it is suitable to focus upon shoulder injuries, when the main focus was performance base. We want to be short and concise in the introduction. We hope the reviewer agrees with our point of view.

*are previous study highlight differences between female and male that could support the choice of your sample? Sex-based differences have been reported in morphological and neuromuscular characteristics of such muscles that could influence also sport injuries but also sport performance. I think that this aspect should be described in order to support your work.

We choose women as subjects since there are not many studies on women compared with men. So that is very interesting. Furthermore, we had access to women for this study which is also an advantage. Since no earlier studies are conducted with either men and women with this wearable resistance it does not matter if we used men or women. In future studies we should use men and see what the effect is of this type of training in men. That is also suggested in future studies.

*" Training throws with a pulley device (heavy loads), throwing overweight balls, and forearm loading resulted in more conflicting results. However, there are some challenges with the different training methods. There are no similar size underweight balls as regular balls; with the over-weighted balls, it is not possible to individualise the weights and throwing with a resistance band and pulley device cannot be conducted during regular throwing practice in  training. Yet, forearm loading is easy to implement during throwing practice during a handball training session, and the load can be easily individually adjusted." references are missing

This is shown in a recent review in which it is referred to in the first sentence. These findings were the result of the review. We have added the reference of the review on the end again, since referring to the different references in the review is not the conclusion of those studies. The conclusion was found in the review.

METHOD

*Inclusion criteria: the range of years is more large had all female examined already had menarche?

All the subjects were over 18 years old and had their period.

*in which phase of menstrual cycle were the participants assessed? Menstrual cycle could modify the sport performance, please specify it.

We think that this is not relevant as recent studies have shown that menstrual cycle did not effect sports performance. Furthermore, It is an intervention period for a team in which we are bounded by the training period and when to test before and after the intervention period. If we would try to test each person in the same part of the cycle, this would not be possible with the training plan and competition of the team. But as said before, recent studies have found that the menstrual cycle did not influence sports performance similarly for most women.  

* 2.4. Measurements. Please add more references that support the chosen instruments of assessment.

We have included some references for measuring the throwing velocity. For the assessment about strength is there only one study that has done the same, which we refer to. The kinematics assessment procedure and instruments are referred by several studies.

RESULTS

*no comment

Thank you.

DISCUSSION

* the results concerning no significant difference in the kinematics and strength test should be better discuss suggesting general hypotheses.

We have discussed from lines 320-359 the differences in kinematics and that no differences in the strength test were found, which were related to the hypotheses. We were also surprised that the strength measurements did not show any significant improvements, which we hypothesized. We hope that the reviewer is informed enough about this.

* the reference to figures and tables in the text should not be placed in the discussions but only in the results, line 308, 318, 323, 327

We think it is better to refer to the figures and tables, but as the reviewer suggested, we have deleted them now from the discussion.

* in general the discussion need a clear statement of the novelties presented in this study

According to us there is a clear statement of the novelties which is mentioned as the conclusion: “Based on findings in this present study, it was concluded that training with wearable resistance attached to the forearm increases throwing velocity during competition season compared to normal throwing training, which was probably caused by the changes in maximal elbow angle and extension velocity and the more flexed wrist at the ball release.”

Reviewer 2 Report

Comments and Suggestions for Authors

Dear Authors,

Thank you for the invitation to review this study. The study looked at assess the effect of a specific training on several parameters in female handball players. Please find some specific comments below.

Introduction

1)Please improve the first paragraph that introduce your research topic. 

Please clarify what you are trying to say, e.g., handball can cause sport injuries that could prevent with exercises..why is it important to improve these kinematics, strength and velocity? Morover the activation of several muscles might contribute to fadigue that might affect the shoulder's strenght proprioception and range of motion, witch in turn could lead to overuse shoulder injury. Take in to consideration the following articles concerning overhead shoulder sport and fadigue:

Buoite Stella A, Cargnel A, Raffini A, Mazzari L, Martini M, Ajčević M, Accardo A, Deodato M, Murena L. Shoulder Tensiomyography and Isometric Strength in Swimmers Before and After a Fatiguing Protocol. J Athl Train. 2024 Jul 1;59(7):738-744. doi: 10.4085/1062-6050-0265.23. PMID: 38014804; PMCID: PMC11277270.

Bauer J, Hagen M, Weisz N, Muehlbauer T. The Influence of Fatigue on Throwing and YBT-UQ Performance in Male Adolescent Handball Players. Front Sports Act Living. 2020 Jul 3;2:81. doi: 10.3389/fspor.2020.00081. PMID: 33345072; PMCID: PMC7739650.

2) the gap in the previous studies should be better descrobed in order to support your study

3) a clearer statement of study's aims would support reader's to understand the study direction.

Method

4) trial registration number should be added

5) The manuscript does not clearly justify the chosen sample size or discuss how it ensures sufficient power to detect expected differences between groups.

6) was the study blinded?

7) the inclusion/exclusion criteria should be better decribed

Results

8) I think that the graphical rappresentation of your data that you have chosen does not cleary illustrate your data, please could you change the graphs? The quality of the tables are very high. 

Discussion

9)The study limitation section should be explain, but also you can  balance this part with your study strengh. Could you balance three limitations with three strenghs? This would provide a more balanced view.

10) the observed benefit of the training with werable resistence should be expanded in order to suggest further investigation and research

Author Response

We want to thank the reviewers for their comments to the manuscript. We have tried to answer to all the comments of the reviewers and changes are colored red in the manuscript. We think that the manuscript now is suitable for publication.

Thank you for the invitation to review this study. The study looked at assess the effect of a specific training on several parameters in female handball players. Please find some specific comments below.

Introduction

1)Please improve the first paragraph that introduce your research topic.

Please clarify what you are trying to say, e.g., handball can cause sport injuries that could prevent with exercises..why is it important to improve these kinematics, strength and velocity? Morover the activation of several muscles might contribute to fadigue that might affect the shoulder's strenght proprioception and range of motion, witch in turn could lead to overuse shoulder injury. Take in to consideration the following articles concerning overhead shoulder sport and fadigue:

Buoite Stella A, Cargnel A, Raffini A, Mazzari L, Martini M, Ajčević M, Accardo A, Deodato M, Murena L. Shoulder Tensiomyography and Isometric Strength in Swimmers Before and After a Fatiguing Protocol. J Athl Train. 2024 Jul 1;59(7):738-744. doi: 10.4085/1062-6050-0265.23. PMID: 38014804; PMCID: PMC11277270.

Bauer J, Hagen M, Weisz N, Muehlbauer T. The Influence of Fatigue on Throwing and YBT-UQ Performance in Male Adolescent Handball Players. Front Sports Act Living. 2020 Jul 3;2:81. doi: 10.3389/fspor.2020.00081. PMID: 33345072; PMCID: PMC7739650.

In the introduction we are not talking about shoulder injuries and only about possibilities of how to enhance throwing performance in handball players. We want to be short and concise and think that it is not interesting to include more information about shoulder injuries. We hope that the reviewer agrees with our point of view.

2) the gap in the previous studies should be better described in order to support your study

The main aim of the present study was to investigate the effect of six weeks of training with wearable resistance attached to the forearm on throwing kinematics, strength, and velocity in female handball players. Since there are only 2 studies that have investigated this with only 0.1 kg weight in novice handball players, not much can be discussed. We have discussed the shortcomings of those studies in lines 60-69 and think that the reader is fully informed about the previous studies and their shortcomings

3) a clearer statement of study's aims would support reader's to understand the study direction.

We have clearly stated in lines 69-73 the studies aim: Therefore, the aim of the present study was to investigate the effect of six weeks of training with wearable resistance attached to the forearm on throwing kinematics, strength, and velocity in female handball players. It was hypothesized that due to the gradual overload of forearm loading, while keeping the same throwing technique, throwing strength will increase, and thereby throwing velocity over time will increase.

We think cleared than this is not possible.

Method

4) trial registration number should be added

It is not a medical trial, so no trial registration should be added. We have the approval date of the ethical approval added to the methods part.

5) The manuscript does not clearly justify the chosen sample size or discuss how it ensures sufficient power to detect expected differences between groups.

In lines 92-95 we have justified the chosen sample size: A minimum sample size of 12 for each group would provide a power of 0.80, calculated by using G*power (Version 3.1.9.6, University of Dusseldorf, Germany) [24]. The power analysis was computed with an effect size of 0.6 and an alpha level of 0.05. There-fore, a sample of more than 12 in each group was assembled. At the start of this study, 30 senior female handball players volunteered. So we think that this justifies the chosen sample size.

6) was the study blinded?

This is very difficult when some wear wearable resistance and other not. So that is not possible.

7) the inclusion/exclusion criteria should be better decribed

We think that we are accurate with our description of the inclusion and exclusion criteria as stated down here:

Inclusion criteria were a) females, 16–40 years old; b) a minimum of five years of handball experience; and c) playing active handball during the study and the last year. Exclusion criteria were a) injuries preventing maximum effort in throwing velocity and b) failure to attend more than 85% of the training sessions. All testing and training were performed in the competitive in-season (September – December, January – March).

Results

8) I think that the graphical rappresentation of your data that you have chosen does not cleary illustrate your data, please could you change the graphs? The quality of the tables are very high.

We think that it is important to show the individual changes in throwing velocity as this is the main outcome that we are interested in. This shows clearly the individual and average changes per group. This you can’t show by a table. That is why we have chosen for this figure. So what exactly wants the reviewer that we could change in the graphs?

Discussion

9)The study limitation section should be explain, but also you can  balance this part with your study strengh. Could you balance three limitations with three strenghs? This would provide a more balanced view.

The limitation section is mainly about limitations. However, we have included some strengths of using the current set up. We hope the reviewer agrees with our strength of the study: However, the strength of this strength test was that it mimics the throwing movement, which is the most specific strength test for overhead throwing (Rivilla-García et al., 2011). Another challenge was recruiting a sufficient number of participants who met the inclusion criteria. The findings are, therefore, applicable only to experienced intermediate-level handball players. The advantage of use of these participants was that both groups were from the same training group and thereby the training content was similar for both except the wearable resistance. It would be valuable to examine forearm loading in elite handball players before the findings are generalized to higher levels. Additionally, the six-week training intervention may have been too short to observe significant strength gains from throwing training with external weights. In the present study, all players followed the same load increase every week, which could influence the results as some players were stronger, taller, and heavier than others. Therefore, the training stimulus could be different for each player. Perhaps individualizing forearm load based on anthropometrics and/or upper body strength could overcome this limitation, which is easy to alter by adjusting the weight of wearable resistance attached to the sleeve during training.

10) the observed benefit of the training with wearable resistence should be expanded in order to suggest further investigation and research

We have under conclusion a part which are practical application for coaches and players, which we think gives many opportunities for them: The advantage of using forearm loading in practice for trainers and players is that it can easily be incorporated throughout an entire training session, including various throwing drills and game sequences as, unlike throwing and shooting with overweighted balls, players can safely shoot at the goalkeeper without increasing the risks of injury. Further-more, the weights can be individualized based on the player’s position and physical ca-pacity, allowing for tailored training programs. This approach enables players to progress at their level while adhering to the principles of progressive overload, ensuring positive adaptations over time.

Under limitations we have suggested some future studies. We hope that this is enough information.

Reviewer 3 Report

Comments and Suggestions for Authors

Many greetings to the authors for their contributions in this field of research. 

Below, you will find my comments on your manuscript. 

General comment 

This is a fascinating study, and its novelty is that it applies to female athletes. However, my main doubt is the unequal groups (14 vs. 11 athletes). They may have affected the study's results. Is this possible? 

Did you control the menstruation cycle of female athletes? 

Specific comments 

Abstract 

L13-15: I think further information about your study design and the experimental variables may be needed. For example, what was the experimental or control group training during the six-week period? 

L16-21: I think the results section should be brief. Include the main findings of your study with the appropriate statistical numbers. 

Introduction 

I think your introduction section is related to the topic of the current manuscript. 

Methods 

L221-233: Please include the statistical power of your study according to your sample size. 

Figure 4. It is wordy. Please revise it. 

Results 

Table 2 and Table 3. The numbers in the tables are too small. There is a need to revise it with better clarity in numbers.

Discussion 

L367-368: include the limitations of your study. 

Conclusions 

I think that you need to include more practical applications. What are the proposals for coaches?  

Author Response

We want to thank the reviewers for their comments to the manuscript. We have tried to answer to all the comments of the reviewers and changes are colored red in the manuscript. We think that the manuscript now is suitable for publication.

Reviewer 3

Many greetings to the authors for their contributions in this field of research.

Below, you will find my comments on your manuscript.

General comment

This is a fascinating study, and its novelty is that it applies to female athletes. However, my main doubt is the unequal groups (14 vs. 11 athletes). They may have affected the study's results. Is this possible?

That is possible, that is also why we have chosen to show the individual changes of the throwing velocity as that is the main outcome we are interested. This gives a good view about that wearable resistance works in general, while just regular training during the season does not have a positive effect upon throwing velocity.

Did you control the menstruation cycle of female athletes?

As mentioned also to the other reviewer, we did not control for menstruation cycle. We think that this is not relevant as recent studies have shown that menstrual cycle did not affect sports performance. Furthermore, It is an intervention period for a team in which we are bounded by the training period and when to test before and after the intervention period. If we would try to test each person in the same part of the cycle, this would not be possible with the training plan and competition of the team. But as said before, recent studies have found that the menstrual cycle did not influence sports performance similarly for most women.  

Specific comments

Abstract

L13-15: I think further information about your study design and the experimental variables may be needed. For example, what was the experimental or control group training during the six-week period?

We have stated in the methods part that under lines 194-213 that both the resistance and the control group did exactly the same training sessions. The only difference was that the wearable resistance training group had the weight attached to the forearms, while the control group did not have that. Thereby the only difference between the groups it this wearable resistance and nothing else. In the abstract this is also mentioned: the same training sessions …

L16-21: I think the results section should be brief. Include the main findings of your study with the appropriate statistical numbers.

We have tried to write the results part briefly. We have shortened it a little bit. We only mention the p values, because otherwise we will get too many numbers in the results part and is the results part not brief anymore.

Introduction

I think your introduction section is related to the topic of the current manuscript.

Thank you

Methods

L221-233: Please include the statistical power of your study according to your sample size.

In lines 92-96 the sample size calculation is done to have a power of at least 0.8, which resulted in a minimum of 12 subjects. This is already mentioned and does not need to be mentioned again later in our opinion.

Figure 4. It is wordy. Please revise it.

In figure 4 the legend is: The forearm sleeves and the attachment of the weights. Adapted from Fredriksen and van den Tillaar [2], which in our opinion is difficult to write shorter. If the reviewer has a suggestion we are open for this.

Results

Table 2 and Table 3. The numbers in the tables are too small. There is a need to revise it with better clarity in numbers.

Table 3 is in 9pt as suggested by the format of the journal. Table 2 have we now split in two tables to get it in the 9pt font for clarity.

Discussion

L367-368: include the limitations of your study.

We think that we have included many limitations of the study. Before these suggestions, we have given 3-4 limitations of the study from line 357 to 374. The other reviewer even suggests some strength of the present study in this limitation part, which we have included now.

Conclusions

I think that you need to include more practical applications. What are the proposals for coaches? 

We think that we already included this under the conclusion part. We have now specified it more for the coaches and players in: The advantage of using forearm loading in practice for coaches and players is that it can easily be incorporated throughout an entire training session, including various throwing drills and game sequences as, unlike throwing and shooting with overweighted balls, players can safely shoot at the goalkeeper without increasing the risks of injury. Further-more, the weights can be individualized based on the player’s position and physical capacity, allowing for tailored training programs. This approach enables players to progress at their level while adhering to the principles of progressive overload, ensuring positive adaptations over time

Round 2

Reviewer 2 Report

Comments and Suggestions for Authors

Dear Author,

thank you again for the invitation to review again your work. 

This updated edited version is improved, but not jet congruent in flow of ideas, and continuity of thought. Unfortunately I can not commend the authors for their efforts, but I can suggest to try again to take in to consideration my suggestions point by point. I will more then happy to read the final version. 

Author Response

We want to thank the reviewer for the comments to the manuscript. We have tried to answer to all the comments of the reviewer and changes are colored red in the manuscript. We think that the manuscript now is suitable for publication.

Thank you for the invitation to review this study. The study looked at assess the effect of a specific training on several parameters in female handball players. Please find some specific comments below.

 Introduction

 1)Please improve the first paragraph that introduce your research topic.

 Please clarify what you are trying to say, e.g., handball can cause sport injuries that could prevent with exercises..why is it important to improve these kinematics, strength and velocity? Morover the activation of several muscles might contribute to fadigue that might affect the shoulder's strenght proprioception and range of motion, witch in turn could lead to overuse shoulder injury. Take in to consideration the following articles concerning overhead shoulder sport and fadigue:

Buoite Stella A, Cargnel A, Raffini A, Mazzari L, Martini M, Ajčević M, Accardo A, Deodato M, Murena L. Shoulder Tensiomyography and Isometric Strength in Swimmers Before and After a Fatiguing Protocol. J Athl Train. 2024 Jul 1;59(7):738-744. doi: 10.4085/1062-6050-0265.23. PMID: 38014804; PMCID: PMC11277270.

Bauer J, Hagen M, Weisz N, Muehlbauer T. The Influence of Fatigue on Throwing and YBT-UQ Performance in Male Adolescent Handball Players. Front Sports Act Living. 2020 Jul 3;2:81. doi: 10.3389/fspor.2020.00081. PMID: 33345072; PMCID: PMC7739650.

In the first paragraph we introduce that throwing is important in several sports and that technique has an effect upon the parameters accuracy and throwing speed. We have included some sentences to make this clearer for the reader. We don’t think that articles about strength in swimmer and fatigue on handball players are suitable references here as we are not talking about fatigue in the article.

2) the gap in the previous studies should be better described in order to support your study

The main aim of the present study was to investigate the effect of six weeks of training with wearable resistance attached to the forearm on throwing kinematics, strength, and velocity in female handball players. Since there are only 2 studies that have investigated this with only 0.1 kg weight in novice handball players, not much can be discussed. We have discussed the shortcomings of those studies in lines 60-69 and think that the reader is fully informed about the previous studies and their shortcomings. So if the reviewer think that there still is a gap, let us know exactly where this gap is.

3) a clearer statement of study's aims would support reader's to understand the study direction.

We have clearly stated in lines 69-73 the studies aim: Therefore, the aim of the present study was to investigate the effect of six weeks of training with wearable resistance attached to the forearm on throwing kinematics, strength, and velocity in female handball players. It was hypothesized that due to the gradual overload of forearm loading, while keeping the same throwing technique, throwing strength will increase, and thereby throwing velocity over time will increase.

We think cleared than this is not possible. If not, please specify what the reviewer means about what is missing.

Method

4) trial registration number should be added

It is not a medical trial, so no trial registration should be added. We have the approval date of the ethical approval added to the methods part.

5) The manuscript does not clearly justify the chosen sample size or discuss how it ensures sufficient power to detect expected differences between groups.

In lines 92-95 we have justified the chosen sample size: A minimum sample size of 12 for each group would provide a power of 0.80, calculated by using G*power (Version 3.1.9.6, University of Dusseldorf, Germany) [24]. The power analysis was computed with an effect size of 0.6 and an alpha level of 0.05. There-fore, a sample of more than 12 in each group was assembled. At the start of this study, 30 senior female handball players volunteered. So we think that this justifies the chosen sample size.

6) was the study blinded?

This is very difficult when some wear wearable resistance and other not. So that is not possible. Since the researcher that tested the players was at all training sessions to control the training sessions and help with putting the wearable resistance on, it was difficult to blind this study.

7) the inclusion/exclusion criteria should be better decribed

We think that we are accurate with our description of the inclusion and exclusion criteria as stated down here:

Inclusion criteria were a) females, 16–40 years old; b) a minimum of five years of handball experience; and c) playing active handball during the study and the last year. Exclusion criteria were a) injuries preventing maximum effort in throwing velocity and b) failure to attend more than 85% of the training sessions. All testing and training were performed in the competitive in-season (September – December, January – March).

Results

8) I think that the graphical rappresentation of your data that you have chosen does not cleary illustrate your data, please could you change the graphs? The quality of the tables are very high.

We think that it is important to show the individual changes in throwing velocity as this is the main outcome that we are interested in. This shows clearly the individual and average changes per group. This you can’t show by a table. That is why we have chosen for this figure. So what exactly wants the reviewer that we could change in the graphs?

Discussion

9)The study limitation section should be explain, but also you can  balance this part with your study strengh. Could you balance three limitations with three strenghs? This would provide a more balanced view.

The limitation section is mainly about limitations. However, we have included some strengths of using the current set up. We hope the reviewer agrees with our strength of the study: However, the strength of this strength test was that it mimics the throwing movement, which is the most specific strength test for overhead throwing (Rivilla-García et al., 2011). Another challenge was recruiting a sufficient number of participants who met the inclusion criteria. The findings are, therefore, applicable only to experienced intermediate-level handball players. The advantage of use of these participants was that both groups were from the same training group and thereby the training content was similar for both except the wearable resistance. It would be valuable to examine forearm loading in elite handball players before the findings are generalized to higher levels. Additionally, the six-week training intervention may have been too short to observe significant strength gains from throwing training with external weights. In the present study, all players followed the same load increase every week, which could influence the results as some players were stronger, taller, and heavier than others. Therefore, the training stimulus could be different for each player. Perhaps individualizing forearm load based on anthropometrics and/or upper body strength could overcome this limitation, which is easy to alter by adjusting the weight of wearable resistance attached to the sleeve during training.

10) the observed benefit of the training with wearable resistence should be expanded in order to suggest further investigation and research

We have under conclusion a part which are practical application for coaches and players, which we think gives many opportunities for them: The advantage of using forearm loading in practice for trainers and players is that it can easily be incorporated throughout an entire training session, including various throwing drills and game sequences as, unlike throwing and shooting with overweighted balls, players can safely shoot at the goalkeeper without increasing the risks of injury. Further-more, the weights can be individualized based on the player’s position and physical capacity, allowing for tailored training programs. This approach enables players to progress at their level while adhering to the principles of progressive overload, ensuring positive adaptations over time.

Under limitations we have suggested some future studies. We hope that this is enough information.

Reviewer 3 Report

Comments and Suggestions for Authors

I want to thank the authors for answering all my comments. I think the manuscript is ready for publication.